# Deep Learning Algorithms for Diagnosis of Lung Cancer: A Systematic Review and Meta-Analysis

**DOI:** 10.3390/cancers14163856

**Published:** 2022-08-09

**Authors:** Gabriele C. Forte, Stephan Altmayer, Ricardo F. Silva, Mariana T. Stefani, Lucas L. Libermann, Cesar C. Cavion, Ali Youssef, Reza Forghani, Jeremy King, Tan-Lucien Mohamed, Rubens G. F. Andrade, Bruno Hochhegger

**Affiliations:** 1Faculty of Medicine, Pontifícia Universidade Católica do Rio Grande do Sul, Porto Alegre 90619-900, Brazil; 2Department of Radiology, Stanford University, Stanford, CA 94205, USA; 3Hospital São Lucas da Pontifícia, Universidade Católica do Rio Grande do Sul, Porto Alegre 90619-900, Brazil; 4Faculty of Medicine, Universidade do Vale do Sinos, Porto Alegre 90470-280, Brazil; 5Radiomics and Augmented Intelligence Laboratory (RAIL), Department of Radiology, University of Florida College of Medicine, Gainesville, FL 32610, USA

**Keywords:** lung cancer, artificial intelligence, deep learning, CNN, deep learning networks

## Abstract

**Simple Summary:**

Lung cancer screening has been shown to help reduce mortality in selected populations of smokers; however, performing screening programs at a larger scale with high accuracy is still a challenge. The use of artificial intelligence (AI) has been investigated to improve large scale screening. We have performed a meta-analysis of the diagnostic accuracy of deep learning (DL) algorithms to diagnose lung cancer. Combining six eligible studies, the pooled sensitivity and specificity of DL algorithms were 0.93 (95% CI 0.85–0.98) and 0.68 (95% CI 0.49–0.84), respectively. Despite remaining challenges in the field, AI is likely to play an important role in disease screening in the future.

**Abstract:**

We conducted a systematic review and meta-analysis of the diagnostic performance of current deep learning algorithms for the diagnosis of lung cancer. We searched major databases up to June 2022 to include studies that used artificial intelligence to diagnose lung cancer, using the histopathological analysis of true positive cases as a reference. The quality of the included studies was assessed independently by two authors based on the revised Quality Assessment of Diagnostic Accuracy Studies. Six studies were included in the analysis. The pooled sensitivity and specificity were 0.93 (95% CI 0.85–0.98) and 0.68 (95% CI 0.49–0.84), respectively. Despite the significantly high heterogeneity for sensitivity (I^2^ = 94%, *p* < 0.01) and specificity (I^2^ = 99%, *p* < 0.01), most of it was attributed to the threshold effect. The pooled SROC curve with a bivariate approach yielded an area under the curve (AUC) of 0.90 (95% CI 0.86 to 0.92). The DOR for the studies was 26.7 (95% CI 19.7–36.2) and heterogeneity was 3% (*p* = 0.40). In this systematic review and meta-analysis, we found that when using the summary point from the SROC, the pooled sensitivity and specificity of DL algorithms for the diagnosis of lung cancer were 93% and 68%, respectively.

## 1. Introduction

Lung cancer has been the leading cause of cancer death for decades [1]. From 2007 to 2017, its incidence has increased by 37% [2], and the number of deaths attributable to lung cancer was over 130 thousand in 2022 only in the United States (US) [1,2]. Due to the asymptomatic nature of early-stage lung cancer, most new cases are diagnosed with advance-stage disease, which often has a poor prognosis with an overall 5-year survival rate of 20.5% [1,3]. From all imaging modalities, computed tomography (CT) is the primary method for the diagnosis and screening of lung cancer given its availability, costs, and optimal spatial resolution of the images [4,5]. Despite the invariable use of ionizing radiation inherited to the technique, it has been shown that low-dose CT (LDCT) for lung cancer screening can be accurately performed with an average effective dose around 1.5 mSv [5,6].

The United States Preventive Services Task Force (USPSTF) recommends lung cancer screening using a LDCT for adults aged 50 to 80 years and have a 20-pack-year smoking history or have quit within the past 15 years [3]. The National Lung Screening Trial (NLST) showed a 20% mortality reduction with screening using LDCT when compared to chest radiography [7]. Additionally, the NELSON trial reported a 26% mortality reduction in men and up to 61% mortality reduction in women with LDCT screening versus no screening [8]. Both trials were critical for the widespread adoption of lung cancer screening strategies in the US and some European Countries [9].

However, there are two main limitations that preclude a more widespread adoption of lung cancer screening programs. One of the concerns is human and technical availability, as radiology capacity may become insufficient to meet the demand [10,11]. The second potential shortcoming is related to false positives cases and overdiagnosis, which is tightly related to the former, given the importance of robust and high-quality training recommended for the providers interpreting the images [10,12]. In previous studies, the benign incidence for a diagnostic operation following nodule discovery was found to be as high as 40% [13,14], highlighting the importance of rigorous nodule screening before more invasive treatments to limit surgical risk and prevent unnecessarily complications or loss of pulmonary capacity.

Considering these limitations, artificial intelligence has been extensively investigated in recent years to be used in computer-aided detection (CAD) systems for the automated detection and/or classification of lung cancer [15]. The effectiveness of the lung cancer screening programs is anticipated to increase with the use of a risk-based tailored strategy and an accurate lung cancer risk prediction model. The ideal CAD would simulate all three steps involved in the analysis of a chest CT for the purposes of lung cancer screening similarly to a radiologist. The first step is the identification of an abnormality in the 3D image set for the presence of one or more regions of interest (ROI), such as a nodular opacity. The second step is to extract all relevant features related to those ROIs, such as dimension, texture, relationship to adjacent areas, among others. Lastly, the extracted features would be used to classify the ROIs according to the likely of malignancy, which is often carried out using validated criteria such as the Lung-RADS [16]. This final step is essential for determining the next step in patient management. In addition, lung segmentation is another important step that CADs often are required to execute for feature extraction, which consist of identifying the voxels of interest of a given ROI. This represents an extra step compared to radiologists who seldom perform 3D segmentation in clinical practice due to time constrains [12].

Recent advances in computational power and deep learning (DL), particularly convolutional neural networks (CNN), resulted in a major shift in the capabilities of CAD, as the performance of non-deep learning algorithms is often below the ideal for clinical application [17], However, most of the literature on CAD has focused on either detection [18,19,20,21], segmentation [22,23,24,25] or classification alone [26,27,28,29,30], which does not decrease the workload of a trained radiologist in clinical practice and thus has hindered the adoption of these methods. More recently, particularly after the Data Science Bowl 2017 (DSB17), many solutions were proposed focused on lung cancer patient diagnosis [31]. Most of the designed solutions consisted of two parts: selecting ROIs through a detection or segmentation module, followed by a malignancy classification module based on the data detected by the previous module [17]. Some studies have proposed end-to-end systems that are able to analyze raw data from CT without the need for segmentation of the ROIs, which can represent a prohibitively time-consuming step, enabling both identification of the areas of interest and classification as malignant or benign [32]. These approaches have shown promising results for the detection and diagnosis of lung cancer [32]. The article by the group from Google is the first large-scale peer-reviewed study to apply deep learning for segmentation and diagnosis using the entire chest CT dataset [33].

We conducted this systematic review and meta-analysis to evaluate the diagnostic performance of current deep learning networks for the diagnosis of lung cancer on CT.

## 2. Materials and Methods

### 2.1. Literature Search

This study was performed using Enhancing the Quality and Transparency of Health Research (EQUATOR) Reporting Guidelines with the Preferred Reporting Items for Systematic Reviews (PRISMA). The study protocol was registered in PROSPERO (CRD 42022347639). A systematic search was conducted in different databases including PubMed (U.S. National Library of Medicine), Embase (Elsevier), and the Scientific Electronic Library Online (Scielo) electronic databases through June 2022. Many publications were identified from reference lists of relevant articles using the “Snowball Method”. Combinations of the equivalent terms were adapted to be used in the search algorithm listed in Appendix A.

### 2.2. Inclusion and Exclusion Criteria

To be included, studies had to meet several criteria: (i) performance evaluation of deep learning for diagnosis of lung cancer; (ii) the validation cohort should have had a histopathological analysis of the true positive cases as a reference; and (iii) data on the true positive (TN), false positive (FP), false negative (FN), and true negative (TN) could be extracted from the manuscript.

Studies were excluded if they (i) performed only the detection of lung cancer and/or nodules; (ii) performed only classification of lung cancer and/or nodules; (iii) reference standard of validation cohort was based on radiologist opinion; (iv) had a sample of fewer than 10 patients; (v) were published as a conference abstract, unrefereed preprints, reviews, or case series.

Two researchers reviewed the titles and abstracts of retrieved articles and applied inclusion and exclusion criteria. The full texts of qualifying articles were retrieved and reviewed to confirm study eligibility.

### 2.3. Assessment of Methodologic Quality

The quality of the included studies was assessed independently by two investigators based on the revised Quality Assessment of Diagnostic Accuracy Studies (QUADAS-2), and all disagreement was resolved through discussion with a third investigator [34]. This quality control instrument consists of four parts: patient selection, index testing, reference standard, and flow and timing. The final criterion is based on the risk of bias with respect to concerns about applicability. Rating risks of bias was determined as high, low, or unclear.

### 2.4. Data Extraction

Literature accepted for analysis was reviewed by two analysts using the PRISMA guidelines [35]. Information collected from studies included first author, year of publication, study design, country of patient recruitment, patient enrollment, technical specifications, reference standard, DL algorithm, and validation cohort. Details regarding the number of TP, TN, FP, and FN were also retrieved from each article. If more than one algorithm was investigated in one study, we would extract data from the algorithm with the highest accuracy. If more than one threshold was investigated, our plan was to extract data from the approach with the highest sensitivity.

### 2.5. Data and Statistical Analysis

Pooled sensitivity and specificity for included studies with a 95% confidence interval (95% CI) were obtained using a random-effects analysis and forest plots were constructed. Summary receiver-operating characteristic curves using the bivariate method were constructed to display the summary point and the area under the curve (AUC) were calculated. The diagnostic odds ratio (DOR) will also be computed with the 95% CI. The inconsistency index (I^2^) was calculated to assess heterogeneity between studies. Given expected heterogeneity between diagnostic accuracies studies due to the inverse relationship between sensitivity and specificity, we have quantified the threshold effect using Spearman’s correlation coefficient between logit sensitivity and logit specificity, and a coefficient (ρ) ≥ −0.6 was considered [36]. The Deeks funnel plot was planned to assess for study asymmetry and potential publication bias if the total number of studies was higher than 10. All analyses were conducted using R (R Project for Statistical Computing).

## 3. Results

### 3.1. Search Results

The initial systematic search identified 3098 studies and an additional 34 records were identified through other sources. After removing of duplicates, 1799 articles were retrieved for title and abstract assessment, and 113 articles were selected for full-text evaluation. One hundred seven articles were excluded based on. Finally, six articles were included in this systematic review and meta-analysis. The flowchart of selection for included studies is demonstrated in Figure 1.

The included studies were published between 2019 and 2022. Out of the six included studies, two were conducted in the USA [37,38], two were conducted in China [39,40], one was conducted in the UK [41], and one in Turkey [42]. Five studies used an external dataset for validation [37,38,40,41,42], while one provided diagnostic performance using cross-validation on the same dataset of training [39]. All six studies used convolutional neural networks as the main artificial intelligence tool and considered histopathological diagnosis as a reference standard for confirming malignant nodules. Regarding validation set sources, two studies used lung cancer screening datasets [37,38]. Details information on the selected studies is summarized in Table 1.

### 3.2. Quality Appraisal

We also evaluated the quality of the studies as well as the risk of bias using the revised QUADAS-2 tool (Appendix A). In the “patient selection” domain, all studies were considered to be at relatively low risk of bias. In the “index test” domain, five studies were at low risk of bias, and one was unclear. In “reference standard”, all studies were regarded as low risk of bias. In addition, in terms of “flow and timing, four studies were scored with a low risk of bias, and two were unclear.

### 3.3. Diagnostic Accuracy and Heterogeneity

Figure 2 and Figure 3 show, respectively, the forest plots for the sensitivities and specificities with the appropriate 95% CI. The pooled sensitivity and specificity were 0.93 (95% CI 0.85–0.98) and 0.68 (95% CI 0.49–0.84), respectively. There was statistically significant heterogeneity for sensitivity (I^2^ = 94%, *p* < 0.01) and specificity (I^2^ = 99%, *p* < 0.01). The pooled SROC curve with the bivariate approach yielded an AUC of 0.90 (95% CI 0.86 to 0.92) (Figure 4). The DOR for the studies was 26.7 (95% CI 19.7–36.2) and heterogeneity was 3% (*p* = 0.40).

The correlation between logit sensitivity and specificity was −0.89, which suggests that most of the heterogeneity can be attributed to the threshold effect. The included studies are plotted close to the summary line in Figure 4, which also demonstrates visually that most of the heterogeneity between studies is related to the threshold effect.

## 4. Discussion

This meta-analysis demonstrates that DL algorithms can achieve good diagnostic performance for the diagnosis of lung cancer on chest CT. As non-invasive method, deep learning models can provide support for radiology clinics by assisting in the early detection and classification of lung cancer, which is critical for early diagnosis leading to effective treatment and improved survival.

The NLST showed that lung cancer screening with the use of low-dose CT resulted in a 20% reduction in mortality from lung cancer [7]. However, reading lung cancer screening CT with high accuracy is not a trivial task even for experienced radiologists, given the three dimensionality of the CT image and all information contained in a scan, which is not restricted to the lung parenchyma. Therefore, it is known that radiologists can fail at cancer detection, which can often be attributed to either fixation or recognition errors [43]. Fixation errors happen when a specialist does not focus enough time to a specific area to detect a possible cancer candidate, which is often associated to stress and fatigue related to the high volume in current practice [44]. On the other side of the spectrum, recognition errors occur mostly when an imaging abnormality is not accurately classified as cancer and is mostly related to radiologist’s level of experience given the wide variety of lung cancer presentation, which encompasses more than just nodular or parenchymal abnormalities [45,46]. In addition, when in close proximity to other structures, such as vessels or pleural, nodules can be often hidden until it outgrows these structures or start having mass effect.

For this reason, a global effort has been made in recent years to find solutions to the issues of nodule identification and nodule malignancy evaluation in the context of cancer screening [37]. LUNA16, for instance, was an open challenge for development and evaluation of algorithms capable of automatically detecting lung nodules [47]. Later on, the DSB17, proposed as part of the Cancer Moonshot initiative, took one step further and challenged communities to develop algorithms that accurately determine when lesions in the lungs are cancerous using a data set of thousands of high-resolution lung scans provided by the National Cancer Institute [48].

Predicting malignancy enables to supplement currently used manual interpretation criteria, such as Lung-RADS, which are only capable of estimating cancer risk by subjective grouping [38]. There are two major types of CAD solutions for lung cancer screening. The first is the computer-aided diagnosis (CAD), that is divided into two components: a Computer-Aided Detection (CADe) module that detects suspicious lung nodules and segments them, and a Computer-Aided Diagnosis (CADx) module that performs both nodule-level assessment and patient-level malignancy classification by analyzing suspicious lesions from CADe. However, there are only a few research papers that propose CADe/CADx, reflecting the challenges when screening detection and classification are associated [15,31,33,38,49,50]. The strengths of the studies included in this review was their effort to design a tool able to offer integrated CADe/CADx analysis at the cost of higher computational power required to perform this task.

Most prior CADe studies typically report only a lesion-level classification performance, which is not comparable to Ardila’s work [38]. In contrast, the latter performs human-independent detection and classification on full volumes. Past non-peer reviewed efforts that have attempted direct, automated malignancy prediction from full volumes using deep learning methods reported AUCs as high as 0.88 [37]. However, these models were primarily trained and tested on smaller portions of the NLST dataset and did not evaluate the use of priors and did not report localization metrics. The disadvantage of Ardila et al. work [38] is that they did not the benchmark their method against state-of-the-art Machine Learning method. In Trajanovsk’s study [37], the performance of the DSB Kaggle competition winners was evaluated in all datasets, allowing a thorough, interesting comparison of the new model’s performance against state-of-the-art approaches [31].

Ozdemir et al. [32] was one of the first that introduced a full deep learning system as an end-to-end automated diagnostic tool to diagnose lung cancer using low-dose CT scans [37]. However, we only reported data on AUC of his model instead of more clinically useful data to derive a 2 × 2 table from his research. Many other authors have been also not provided enough information for us to obtain the 2 × 2 data to include in this meta-analysis. Huang et al. [33] developed a deep learning algorithm that accounted for all relevant nodule and non-nodule features on a screening chest CT and demonstrated high accuracy in predicting lung cancer presence over a three-year period, while also generalizable to an external dataset.

Ardila et al. compared the performance of the Lung-RADS classification by six radiologists thresholding the model’s prediction at three different cutoffs [38]. The group showed that the average performance of the radiologists using Lung-RADS to predict lung cancer was at the same level of the DL algorithm. However, there are two benefits of incorporating DL algorithms to clinical practice: ability to meet the increasing demand of higher volumes with performance at the level of an experience radiologist, as well as the inter-reproducibility of the findings to guarantee consistency, especially on patient follow-up. Given the capabilities of more recent algorithms to both identify regions of interest and provide a malignancy estimation, it could theatrically decrease the workload of the radiologist in clinical practice.

Although the performance of DL algorithms is promising in these early investigations, there are still important developments and optimizations required before successful clinical adoption of these tools. For widespread adoption, it is important to demonstrate that the algorithms perform reliably in different real-world settings, including validation of performance on the wide variety of technical acquisitions that may be encountered in clinical practice based on differences in scanner types and technical parameters in scan acquisition and reconstruction. For effective adoption, it is also important to ensure that the systems are compatible and can be seamless integrated into routine clinical practice and the radiology reporting systems, ideally with a standardized set of output parameters such as common data elements or other similar approaches. This includes a consideration of the computational processing requirements and any necessary upgrades in the informatics infrastructure of the enterprise that would ensure effective algorithm adoptions. Once these requirements are met and are the systems can be adopted clinically, it would be important to demonstrate and confirm the positive impact of these tools on patient management and outcomes in prospective studies.

One of the limitations of widespread adoption is the inherent lack of explanation behind the decision of DL models is a great barrier for most radiologists and clinicians who would want to understand the features used by the algorithms to predict malignancy. This is particularly true for end-to-end decision systems; CADs should be able to transmit the reasoning behind the decision so the radiologist and clinician can trust in the decision tools [17]. Another main limitation with most of the more recent DL algorithms is the computational power required for the segmentation of raw data from a CT scan and how long it would take with a conventional machine. In addition, future research endeavors should attempt to include active participation of radiologists in framing the solution to increase the clinical applicability of it. For instance, many papers found in our review provided only the AUC or accuracy of their models, instead of more relevant clinical data such as sensibility, specificity, and likelihood ratios. As the field develops, identification of what is most relevant for clinical practice becomes essential [17]. Lastly, the studies herein included had their data derived from lung cancer screening in a targeted population of smokers, therefore the applicability of our results in the screening of non-smoker Asian populations [51].

## 5. Conclusions

In conclusion, in this systematic review and meta-analysis, we found that the using the summary point from the SROC, the pooled sensitivity and specificity of DL networks for the diagnosis of lung cancer were 93% and 68%, respectively. Despite many improvements still to be made in the field of artificial intelligence in lung cancer detection, the currently available data is promising, and DL based CAD tools are likely to play an important role in lung cancer screening in the near future.

## Figures and Tables

**Figure 1 cancers-14-03856-f001:**
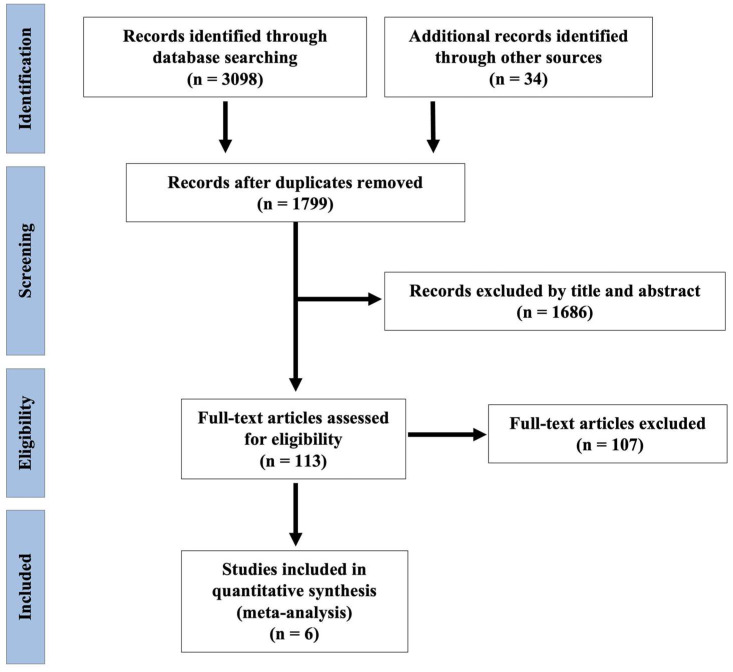
Preferred Reporting Items for Systematic Reviews and Meta-Analyses (PRISMA) flow diagram.

**Figure 2 cancers-14-03856-f002:**
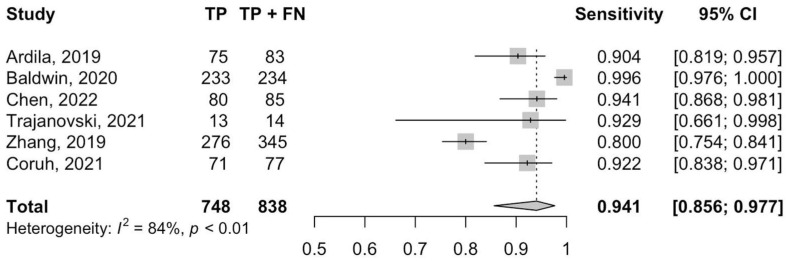
Forest plot of the pooled sensitivity of DL in the detection and classification of lung cancer [37,38,39,40,41,42].

**Figure 3 cancers-14-03856-f003:**
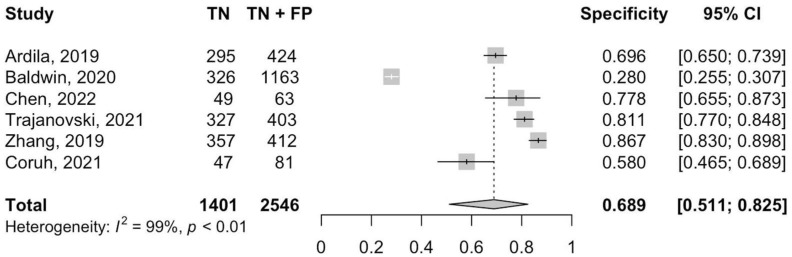
Forest plot of the pooled specificity of DL in the detection and classification of lung cancer [37,38,39,40,41,42].

**Figure 4 cancers-14-03856-f004:**
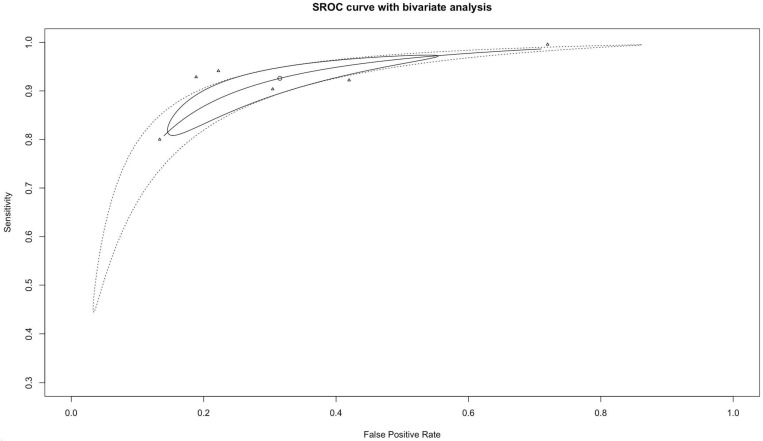
Summarized receiver-operating curves (SROC) using the bivariate approach.

**Table 1 cancers-14-03856-t001:** Characteristics of the included studies.

Author	Year	Country	Study Design	Center	Artificial Intelligence	Source Validation Set	Threshold	Reference Standard Validation	Method Validation
Ardila et al. [38]	2019	USA	retrospective	multicenter	CNN	Lung cancer screening dataset	PPV = 0.11	Histopathology	External validation
Baldwin et al. [41]	2020	UK	retrospective	multicenter	CNN	Private dataset	FN rate = 0%	Histopathology	External validation
Chen et al. [40]	2022	China	retrospective	single	CNN	Private dataset	Unknown (third party software)	Histopathology	External validation
Çoruh et al. [42]	2021	Turkey	retrospective	single	CNN	Private dataset	Youden index optimal cutoff	Histopathology	External validation
Trajanovski et al. [37]	2021	USA	retrospective	multicenter	CNN	Lung cancer screening dataset	Sensitivity = 93%	Histopathology	External validation
Zhang et al. [39]	2019	China	retrospective	multicenter	CNN	Private dataset	Probability of malignancy > 0.5	Histopathology	Cross-validation

CNN: convolutional neural network; PPV: positive predictive value; FN: false negative.

## Data Availability

All data available within the manuscript.

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
