# Peer review of "Deep Learning Algorithms for Diagnosis of Lung Cancer: A Systematic Review and Meta-Analysis"

_cancers, 2022, doi:10.3390/cancers14163856_

Round 1
Reviewer 1 Report
This study tries to address the Deep learning algorithms for diagnosis of lung cancer by systemic review approach. The pool study results show high sensitivity, but fair specificity performance in diagnosis of lung cancer subjects.
Introduction
Please add several sentences to address non-smokers lung cancers & diagnostic performance in Asian population.
Assessment of selection criteria for low-dose lung screening CT among Asian ethnic groups in Taiwan: from mass screening to specific risk-based screening for non-smoker lung cancer
FZ Wu, YL Huang, CC Wu, EK Tang, CS Chen, GY Mar, Y Yen, MT Wu
Clinical lung cancer 17 (5), e45-e56
Method:
Please try to classify the vendors/copay in these studies.
Result and conclusion
Please tries to compare the performance of deep learning software and lung-RADs.
Please try to discuss about the impact of clinical practice of deep learning algorithm on radiologists’ performance and how to deal with extra-loading due to low specify accuracy.
4.Grammar and spelling
A few minor typos, grammar hiccups should also be corrected, e.g., “vanced-stage” in line 42, page 1.
As suggested by the reviewer, we have gone through the revised article and corrected minor typing mistakes and grammatical errors.
Page 1 -Corrected grammatical mistake in the “vanced-stage” to “vance-stage” (line 42)
Page 2 -Corrected grammatical mistake in the “lung cancer” to “the lung cancer” (line 49)
Page 2 -Corrected grammatical mistake in the “lung cancer” to “the lung cancer” (line 69)
Page 2 -Corrected grammatical mistake in the “ideal” to “the ideal” (line 86)
Page 2 -Corrected grammatical mistake in the “have” to “has” (line 87)
Page 2 -Corrected grammatical mistake in the “workload” to “the workload” (line 88)
Page 2 -Corrected mistake in the “these method” to “this method” (line 89)
Page 5 -Corrected grammatical mistake in the “studies” to “study” (line 176)
Page 5 -Corrected grammatical mistake in the “bivariate” to “the bivariate” (line 192)
Page 7 -Corrected grammatical mistake in the “mass” to “the mass” (line 227)
Page 7 -Corrected mistake in the “Computer Aided” to “Computer-Aided” (line 240)
Page 7 -Corrected grammatical mistake in the “are” to “is” (line 244)
Page 7 -Corrected grammatical mistake in the “benchmark” to “the benchmark” (line 253)
Page 8 -Corrected grammatical mistake in the “was” to “were” (line 255)
Page 8 -Corrected grammatical mistake in the “have also not provided” to “have been also not provided” (line 260)
Page 8 -Corrected mistake in the “real world settings” to “real-world settings” (line 269)
Page 8 -Corrected grammatical mistake in the “was” to “were” (line 297)
Author Response
- Please add several sentences to address non-smokers lung cancers & diagnostic performance in Asian population. Assessment of selection criteria for low-dose lung screening CT among Asian ethnic groups in Taiwan: from mass screening to specific risk-based screening for non-smoker lung cancer FZ Wu, YL Huang, CC Wu, EK Tang, CS Chen, GY Mar, Y Yen, MT Wu Clinical lung cancer 17 (5), e45-e56
Answer: We appreciate the Reviewer’s suggestion. We have added this as a limitation in our discussion that we did not assess the population of nonsmokers as all studies included were derived from lung cancer screening studies, therefore the populations were all smokers.
“Lastly, the studies herein included had their data derived from lung cancer screening in a targeted population of smokers, therefore the applicability of our results in the screening of non-smoker Asian populations [51].”
2 Method: Please try to classify the vendors/copay in these studies.
Answer: We thank the Reviewer for the question. All studies have used at least one of the large screening datasets (NSLT, LUNA16, etc) that were multicenter studies using multiple vendors and this information is not readily available. Overall, the data can be seen as an average of a wide variety of vendors for data training and/or validation.
3 Result and conclusion: Please tries to compare the performance of deep learning software and lung-RADs. Please try to discuss about the impact of clinical practice of deep learning algorithm on radiologists’ performance and how to deal with extra-loading due to low specify accuracy.
Answer: We think this is an excellent question. We have incorporated a new parapgrah into our discussion to discuss this issue and explore some shortcoming for DL algorithms to be implemented in clinical practice as below.
Ardila et al. compared the performance of the Lung-RADS classification by 6 radiologists thresholding the model’s prediction at three different cutoffs [38]. The group showed that the average performance of the radiologists using Lung-RADS to predict lung cancer was at the same level of the DL algorithm. However, there are two benefits of incorporating DL algorithms to clinical practice: ability to meet the increasing demand of higher volumes with performance at the level of an experience radiologist, as well as the inter-reproducibility of the findings to guarantee consistency, especially on patient follow-up. Given the capabilities of more recent algorithms to both identify regions of interest and provide a malignancy estimation, it could theatrically decrease the workload of the radiologist in clinical practice.
Although the performance of DL algorithms is promising in these early investigations, there are still important developments and optimizations required before successful clinical adoption of these tools. For widespread adoption, it is important to demonstrate that the algorithms perform reliably in different real-world settings, including validation of performance on the wide variety of technical acquisitions that may be encountered in clinical practice based on differences in scanner types and technical parameters in scan acquisition and reconstruction. For effective adoption, it is also important to ensure that the systems are compatible and can be seamless integrated into routine clinical practice and the radiology reporting systems, ideally with a standardized set of output parameters such as common data elements or other similar approaches. This includes a consideration of the computational processing requirements and any necessary upgrades in the informatics infrastructure of the enterprise that would ensure effective algorithm adoptions. Once these requirements are met and are the systems can be adopted clinically, it would be important to demonstrate and confirm the positive impact of these tools on patient management and outcomes in prospective studies.
4.Grammar and spelling
A few minor typos, grammar hiccups should also be corrected, e.g., “vanced-stage” in line 42, page 1.
Page 1 -Corrected grammatical mistake in the “vanced-stage” to “vance-stage” (line 42)
Page 2 -Corrected grammatical mistake in the “lung cancer” to “the lung cancer” (line 49)
Page 2 -Corrected grammatical mistake in the “lung cancer” to “the lung cancer” (line 69)
Page 2 -Corrected grammatical mistake in the “ideal” to “the ideal” (line 86)
Page 2 -Corrected grammatical mistake in the “have” to “has” (line 87)
Page 2 -Corrected grammatical mistake in the “workload” to “the workload” (line 88)
Page 2 -Corrected mistake in the “these method” to “this method” (line 89)
Page 5 -Corrected grammatical mistake in the “studies” to “study” (line 176). The sentences refers to more than one study.
Page 5 -Corrected grammatical mistake in the “bivariate” to “the bivariate” (line 192)
Page 7 -Corrected grammatical mistake in the “mass” to “the mass” (line 227)
Page 7 -Corrected mistake in the “Computer Aided” to “Computer-Aided” (line 240)
Page 7 -Corrected grammatical mistake in the “are” to “is” (line 244)
Page 7 -Corrected grammatical mistake in the “benchmark” to “the benchmark” (line 253)
Page 8 -Corrected grammatical mistake in the “was” to “were” (line 255)
Page 8 -Corrected grammatical mistake in the “have also not provided” to “have been also not provided” (line 260)
Page 8 -Corrected mistake in the “real world settings” to “real-world settings” (line 269)
Page 8 -Corrected grammatical mistake in the “was” to “were” (line 297)
Answer: We thank the Reviewer for the suggestions, we have modified the text to include the recommendations.
Reviewer 2 Report
The paper provides a literature review of the applications of deep learning methods for lung cancer using the histopathological analysis of true positive cases as a reference.
------------------ strengths --------------------------
- The inclusion criteria of the literature review seem to be thorough.
- The discussion section is well organized
- The meta-analysis of the reviewed papers is complete and informative.
- The future directions and current open issues are highlighted in this section.
- The conclusion section summarized the authors' viewpoints on the provided reviewed papers.
------------------ limitation --------------------------
- What are the benefits and limitations of the utilized studies for lung cancer diagnosis?
- The authors in section 2.3 (Assessment of methodologic quality) highlighted, "Disagreements between reviewers were resolved by consensus" it is suggested to provide more information about how this consensus is reached.
Author Response
The paper provides a literature review of the applications of deep learning methods for lung cancer using the histopathological analysis of true positive cases as a reference.
------------------ strengths --------------------------
- The inclusion criteria of the literature review seem to be thorough.
- The discussion section is well organized
- The meta-analysis of the reviewed papers is complete and informative.
- The future directions and current open issues are highlighted in this section.
- The conclusion section summarized the authors' viewpoints on the provided reviewed papers.
------------------ limitation --------------------------
- What are the benefits and limitations of the utilized studies for lung cancer diagnosis?
Answer: We thank the Reviewer for the question. The main strength of these studies is that fact most performed end-to-end analysis with both identification and classification (CADe/CADx) compared to older studies that only assessed identification or classification separately. However, performing end-to-end analysis, particularly using a full volume from a CT, such as Ardila, is the computational power required to do so. We have introduced these ideas on the introduction as well as discussion, but we have highlighted this point in one more sentence as shown below.
“The strengths of the studies included in this review was their effort to design a tool able to offer integrated CADe/CADx analysis at the cost of higher computational power required to perform this task.”
The authors in section 2.3 (Assessment of methodologic quality) highlighted, "Disagreements between reviewers were resolved by consensus" it is suggested to provide more information about how this consensus is reached.
Answer: We thank the Reviewer for the question. That sentence was excluded from the manuscript as it was inaccurate. It was stated in line 132 (just above this excluded sentence) that “all disagreement was resolved through discussion with a third investigator”, in which we have a system of 2 x 1 votes to reach final veredict..
Reviewer 3 Report
an excellent work congrats.The A.I is something relative new but well accepted and used in daily routine,at least in research field.in 2.2 line 117 consider the possibility to take under consideration the different histological types of cancer:for example sarcomas are not "captured" in CT as the NSCLC,do we have there a flutuance of statistics(sensitivity)?or the mucinus,in situ or minimally invasive adenocarcinomas ,that for certain need different CT diagnostic approach, are these considered?
Author Response
- An excellent work congrats. The A.I is something relative new but well accepted and used in daily routine, at least in research field.in 2.2 line 117 consider the possibility to take under consideration the different histological types of cancer:for example sarcomas are not "captured" in CT as the NSCLC,do we have there a flutuance of statistics(sensitivity)?or the mucinus,in situ or minimally invasive adenocarcinomas ,that for certain need different CT diagnostic approach, are these considered?
Answer: We thank the Reviewer for this excellent question. We agree that most likely each histological type of lung cancer would have a specific diagnostic detection by DL algorithms on LDCT. However, none of the studies included provided data of diagnostic performance per histological type, therefore this data is not available for discussion.